Demographic compensation occurs in populations of Quercus oleoides Schltdl & Cham in fragments across an altitudinal gradient

Flores-Romero Carlos 1 2
http://orcid.org/0000-0002-6014-8731 Sánchez-Velásquez Lázaro Rafael 1 lasanchez@uv.mx
http://orcid.org/0000-0001-5306-7397 Equihua Miguel 2
Pineda López María del Rosario 3
Alarcón Gutiérrez Enrique 1
Perroni Yareni 1
1 Institute of Biotechnology and Applied Ecology (INBIOTECA), Universidad Veracruzana , Xalapa, Veracruz , Mexico
2 Environment and Sustainability Network, Institute of Ecology AC (Inecol) , Xalapa, Veracruz , Mexico
3 Center of EcoAlfabetización y Dialogue of Saberes, Universidad Veracruzana , Xalapa, Veracruz , Mexico
Wehenkel Christian
Electronic publication date: 2025 Feb 24
Publication date: 2025
Volume: 13
Electronic Location ID: e18980
Received 2024 Feb 6; Accepted 2025 Jan 22
Copyright: © 2025 Flores-Romero et al.
Copyright year: 2025
Copyright holder: Flores-Romero et al.
License: This is an open access article distributed under the terms of the Creative Commons Attribution License, which permits unrestricted use, distribution, reproduction and adaptation in any medium and for any purpose provided that it is properly attributed. For attribution, the original author(s), title, publication source (PeerJ) and either DOI or URL of the article must be cited.
License URL: https://creativecommons.org/licenses/by/4.0/

Keywords: Autoregulation, Demography, Elasticity matrix, LTRE variation, Mexico, Oak, Population dynamic, Population growth rate

Funding: Conahcyt Mexico 692814 Carlos Flores-Romero received a doctoral fellowship (692814) from Conahcyt Mexico. The funders had no role in study design, data collection and analysis, decision to publish, or preparation of the manuscript.

==============================
Background

Demographic compensation is a complex process by which populations can compensate for the effects of anthropogenic disturbance and other environmental changes and restore growth-rate stability (λ » 1). Dynamic equilibrium is achieved when the growth rate [λ] is close to one. This enables a population to persist under changing environmental conditions. The demographics of fragmented populations provides an ideal model to explore the processes by which populations adapt through demographic compensation responses.

Aims

To characterize the demographic of Quercus oleoides populations and detect the various processes that result from demographic compensation responses.

Methods

We established permanent plots in three Q. oleoides populations at which three annual transition stages were registered. These were survival probability, transition probability, and average reproduction (that is, the number of seed production by size class).

Results

The λs of the three populations under study were >1.0 (P < 0.005). However, differences were found in the elasticity matrices, as well as in the life-table response experiment (LTRE) variation analyses.

Conclusions

The three Q. oleoides populations have shown to have affected a transition to demographic compensation and achieved demographic balance through different strategies in their vital rates.

Introduction

A species’ wide distribution indicates that it has the ability to adapt to varying local conditions. This is possible when the species has a remarkable phenotypic plasticity or when it undergoes a wide genetic variation (Miner et al., 2005; de Villemereuil et al., 2018; Gurung et al., 2019; Cruzado-Vargas et al., 2021; Figueroa-Rangel & Olvera-Vargas, 2022; Khatri et al., 2023a).

Altitude and latitude gradients are known to be associated mostly to climatic parameters, temperature and precipitation (Aparecido et al., 2018; Arriaga et al., 2000; Sánchez-Velásquez et al., 2023). Plants have the ability to modify their morphophysiological, biochemical, phenological and behavioural traits under changing environmental conditions (Nock, Vogt & Beisner, 2016; Khatri et al., 2022) and exhibit plastic responses to them (Karban, 2015) with remarkable flexibility (Khatri et al., 2023b). During these processes, some characteristics are favoured in order to improve the plant’s performance (Manral et al., 2022; Pandey et al., 2024). When due to disturbance a population gets fragmented along an environmental gradient, its structure and dynamics may change. This will result in population fragments having different population growth and vital rates (Cosset, Gilroy & Edwards, 2019; Caughlin et al., 2019; Nordstrom, Dykstra & Wagenius, 2021; Körner, 2007). However, in monospecific as well as multispecific communities, there are cases in which population growth rates differentiation has not occurred thanks to demographic compensation (Villellas et al., 2015; Yang et al., 2022), resilience, autogenic regulation (Muñoz et al., 2021) or autopoiesis (Luisi, 2003; Froese et al., 2023; Mackey, Morgan & Keith, 2023).

Resilience, a concept widely used in complex ecological systems, can be defined as a system’s ability to resist and recover after disturbance (Holling, 1973). Specifically, demographic resilience can be defined as the ability of populations to resist and recover from alterations in their demographic structure and size (Capdevila et al., 2020).

Demographic compensation occurs when vital rates change in opposite directions among populations affecting variation in population growth rates. This mechanism can explain the stability of the geographic distribution of species under climate change or along an environmental gradient (Andrello et al., 2020; Yang et al., 2022). Demographic compensation is a complex process by which, when faced with environmental changes, populations achieve structural changes by modifying their vital rates to compensate and direct their growth rate towards stability (λ ≈ 1). We assume that demographic compensation can be observed in the changes brought about in the vital rate, stability, sensitivity, and LTRE (life-table response experiment; variation) rates (Tuljapurkar, 1990).

In the study of population dynamics, matrix models are useful theoretical and analytical tools to calculate the vital rates and their behaviour under changing environmental conditions or disturbance (Menges & Dolan, 1998; Caswell, 2001; Morris & Doak, 2002; Sánchez-Velásquez & Pineda-López, 2010; Sánchez-Velásquez et al., 2021; Fortini et al., 2022). The asymptotic population growth rate (λ) summarizes all the survival, growth, and birth rates of a population over time. Other important vital rates are as sensitivity and elasticity. Transition matrices help to calculate all these vital rates as well as LTRE variations (Silvertown, Franco & McConway, 1992; Ehrlén & Van Groenendael, 1998; Menges & Dolan, 1998; Caswell, 2001; Mills & Lindberg, 2002; Morris & Doak, 2002; Franco & Silvertown, 2004). Knowledge of the vital rates allows us to determine whether a population has reached stability through in the context of management practices, that is, whether or not the λs are close to balance (λ ≈ 1).

The fragmentation of communities and populations is a common phenomenon in mountainous areas (Muñoz Gómez, Pérez & Galicia Sarmiento, 2018; Espinoza-Guzmán et al., 2020; Leija et al., 2021). Mexico stands out globally for the concentration of species of the Quercus genus in its territory. Though these species are present at all altitudes, they tend to concentrate in the mountains (Valencia, 2004) up to an altitude of 3,650 m a.s.l. (Gómez-Pompa, 1978). However, the species of the Quercus genus prosper mostly in warm and temperate areas (Gómez-Pompa, 1978). With a total of 150 species, Mexico is one of two diversification centres of the Quercus genus (Valencia, 2004; Hipp et al., 2017). Between 74 and 109 of the species of this genus are endemic (Valencia, 2004; Ricker et al., 2016). Despite its wide distribution from the state of Tamaulipas in Northeastern Mexico down to Costa Rica (Fournier, 2003; Romero-Rangel, Rojas-Zenteno & Rubio-Licona, 2015; Pérez-Mojica & Valencia, 2017), Quercus oleoides is an oak on the IUCN Red List of Engendered Species (Eastwood & Oldfield, 2007).

In Mexico, Q. oleoides is mostly distributed along the Gulf of Mexico floristic province (Morrone, 2001). As refuge zones, Q. oleoides populations provide shade for many plant and animal species during the dry season (Boucher, 1983; Janzen, 1983), helping to preserve a temperature and water balance, storing carbon, and ensuring seasonal biomass productivity (Powers et al., 2009; Kissing & Powers, 2010). Today, most Q. oleoides forests are fragmented and their genetic diversity is in constant decrease due to human activity (Pennington & Sarukhán, 2005).

An investigation of the demographics of Q. oleoides populations in fragments under different environmental conditions and disturbance histories along an altitudinal gradient will yield enough data to evaluate the current state of these populations observe how demographic compensation responds to fragmentation, and detect vulnerability stages and transition variations, so as to devise effective conservation strategies.

Our main concern here is to understand how the vital rates of populations along an altitudinal gradient with different environmental conditions vary. The research questions are: (1) Do the vital rates of Q. oleoides populations vary at different altitudes? (2) Are asymptotic population growth rates the same from one population to another? We hypothesized that if the populations located at different altitudes and with different degrees of disturbance undergo demographic compensation, they would have asymptotic population growth rates (λ ≥ 1) despite their differences in vital rates (such as LTRE and elasticity), which demonstrates their demographic compensation capabilities (Villellas et al., 2015; Yang et al., 2022). However, if asymptotic population growth rates are different and less than one, this means that the populations are not engaged in compensation.

Materials and Methods

A description of Quercus oleoides

Commonly known as tropical or white oak, Q. oleoides is a semi-deciduous tree reaching up to 15 m in height. Its outer bark is pale to dark grey and presents deep fissures 20 to 30 mm thick. Its seed (acorn) is recalcitrant, equipped to keep the necessary humidity to be viable and it doesn’t exhibit dormancy (Klemens, Deacon & Cavender-Bares, 2010). The tree blooms between the months of April and June and the production of seeds takes place between October and November. It is a monoecious species whose flowers are pollinated by the wind, though the movement of its seeds is known to be limited to very short distances due to the poor dispersion behaviour of their main dispersers (Boucher, 1983). It inhabits seasonally dry tropical forests in Mesoamerica, where it is the dominant species. It grows in poor, rocky soils (Boucher, 1983). In general, oaks do not have a persistent seed bank (Olson, 1974).

Study area

The study was conducted in the Priority Terrestrial Region (RTP) 104 “Tropical Oak Woods of the Coastal Plains of Veracruz”, which extends over an area of 905 km2 at the centre of the state between latitudes 19°20.0 to 19°45.0 N, and longitudes 96°30.0 to 97°00.0 W (Table 1) (Arriaga et al., 2000). The climate is warm subhumid with a mean annual temperature above 22 °C and a coldest-month temperature above 18 °C, with an annual rainfall of between 500 and 2,500 mm, ranging from 0 to 60 mm in the driest month. Summer rainfalls represent from 5% to 10.2% of the annual total (Arriaga et al., 2000).

Table 1 General characteristics of the localities where the permanent sites were established for the study of the demography of Quercus oleoides in Veracruz, Mexico.

Locality name	Geographical location	Altitude (m a.s.l.)	Mean no. of individuals/ha (>0 cm dbh) ± Sx	Disturbance level	Fragment Size (ha)	
Miramar	19°40.965 N 96°25.650 W	60	1,345 ± 24	Moderate cattle ranging 4 cow/ha	100	
Mesa de Veinticuatro	19°43.543 N 96°29.919 W	400	2,591.7 ± 34	No cattle ranging 0 cow/ha	96	
Otates	19°30.637 N 96°43.211 W	800	1,831.6 ± 37	Intense cattle ranging 6 cow/ha	23	
Note:

dbh, diameter at breast height (cm); ± Sx, mean standard deviation.

Different disturbance histories correspond with different evolutionary and adaptational pressures. The forest has been fragmented, giving way to isolated populations along the environmental gradient (Pennington & Sarukhán, 2005). Our study site includes these isolated populations. Extensive cattle raising, a common activity in forests dominated by Q. oleoides (Carlos Flores-Romero, 2019, personal observation), has been considered an intervening factor in the selection and transformation of plant populations and communities (Milchunas & Laurenroth, 1993).

Fieldwork data on the survival, transition, and reproduction of Quercus oleoides

In order to analyse the altitudinal range in which Q. oleoides forest occurs (0 to 840 m a.s.l.), we selected three populations resulting from forest fragmentation at different altitudes and with different disturbance histories (Table 1). The first one is located at 60 m a.s.l. in the locality of Miramar in the municipality of Actopan, Ver., with moderate cattle grazing during the rainy season. The second one at 400 m a.s.l, in the locality of Mesa de Veinticuatro in the municipality of Alto Lucero, Ver., without cattle grazing in the last 15 years, and the third population at 800 m a.s.l. in the locality of Otates in the municipality of Actopan, Ver., with more intense cattle grazing combined with some lime plantations (Table 1). The intensity of cattle grazing did not change during the study.

At each one of these altitudinal levels, we established five random square plots of 400 m2 (20 × 20 m) (at 100–200 m from each other, total 15 plots) and within each one of them, all Q. oleoides individuals were registered each year. For those being 1.3 m in stature or shorter, we registered their total height with a measuring tape flexible (Trupper), while for the rest of individuals (i.e., those higher than 1.3 m) the diameter at breast height (dbh) was measured and registered each year with a diameter tape (Forestry Suppliers). We determined the geographical location of each individual with a Garmin GPS receiver (GpSmap 60CSx) and to facilitate their annual location and further registration each one was identified with an aluminium tag bearing a reference number.

Based on the annual post-breeding censuses, we calculated each year the survival and transition and reproduction for the size classes >1.3 m height, whereas for smaller individuals the seed-to-plant transition was estimated by dividing the number of new individuals <1.30 m height into the number of seeds from the previous period. We counted in each reproductive tree (November) the seeds on a randomly selected in the bottom, middle and top tree crown sections using a set of binoculars and a tally counter clicker. Also, we counted the number of branches in each section to calculate the total number of acorns per tree. We conducted the registers during four years to obtain three annual transition matrices per population (2016–2017, 2017–2018, and 2018–2019).

Data analysis

Annual transition matrices and asymptotic population growth rates

In order to determine the population vital rates per altitude (combining the five plots from each), we divided each of the three populations into six size classes: seed stage or acorn, <1.3 m height, >0 ≤ 10 cm dbh, >10 ≤ 20 cm dbh, >20 to ≤ 30 cm dbh, and >30 cm dbh (Fig. 1). We used the transition probabilities among size classes (Pij, number of individuals in size i at time t + 1 divided per number of individuals in size j at time t), survival probability (Gij, number of individuals that remained in size i at time t + 1 divided per number of individuals in size i at time t) and the average seed production by size class (Fij is the average number of seeds produced by individuals in size i from time steps t to t + 1) to estimate the annual transition matrix and λ (the asymptotic population growth rate) (Caswell, 2001) (Fig. 1, Table 2). Therefore, we calculated the three transition matrices, and three average transition matrices for each population (Lewontin & Cohen, 1969; Caswell, 2001). We calculated λ from each of the three annual asymptotic transition matrices by an iterative matrix multiplication process (Caswell, 2001).

Figure 1 Quercus oleiodes’s annual transition per SS.

Pi are the proportion of transition of individuals, Gi are the proportion of individuals that remain, Fi are the average number of seeds per tree. d, diameter at breast height (cm); h, high (m); SS, stage size.

Table 2 Average of transition matrices across three years of observations and its population growth rate (λ) of Quercus oleoide population from Miramar, Mesa de Veinticuatro and Otates localities, Veracruz, Mexico.

Size	Seeds	<1.3 h	>0–10 d	>10–20 d	>20–30 d	>30 d	
Miramar λ¯ = 1.076 (cl = [1.0759–1.0761])	
Seeds	0	0	295.3 ± 138	3,000 ± 1,156	8,000 ± 1,156	7,833.3 ± 727	
<1.3	3.82E−05 ± 2.1E−05	0.158 ± 0.0514	0	0	0	0	
>0–10	0	0.444 ± 0.0241	0.759 ± 0.0110	0	0	0	
>10–20	0	0	0.165 ± 0.0307	0.940 ± 0.1013	0	0	
>20–30	0	0	0	0.022 ± 0.0973	0.307 ± 0.0882	0	
>30	0	0	0	0	0.068 ± 0.0804	0.986 ± 0.0003	
Mesa de Veinticuatro λ¯ = 1.017 (cl = [1.0059–1.0061])	
Seeds	0	0	931.2 ± 808	1,966 ± 607	5,000 ± 1,529	5,333 ± 1,858	
<1.3	1.07E−05 ± 4.3E−06	0.069 ± 0.0231	0	0	0	0	
>0–10	0	0.734 ± 0.0551	0.587 ± 0.1789	0	0	0	
>10–20	0	0	0.281 ± 0.2033	0.648 ± 0.0934	0	0	
>20–30	0	0	0	0.333 ± 0.0946	0.816 ± 0.0727	0	
>30	0	0	0	0	0.174 ± 0.0735	0.988 ± 0	
Otates λ¯ = 1.042 (cl = [1.0223–1.0242])	
Seeds	0	0	745 ± 343	4,500 ± 2,023	4,667 ± 2,188	2,500 ± 1,260	
<1.3	4.33E−05 ± 1.5E−05	0.333 ± 0.1882	0	0	0	0	
>0–10	0	0.541 ± 0.1943	0.761 ± 0.0399	0	0	0	
<10–20	0	0	0.165 ± 0.0093	0.939 ± 0.0107	0	0	
<20–30	0	0	0	0.029 ± 0.0139	0.600 ± 0.2840	0	
>30	0	0	0	0	0.069 ± 0.0108	0.986 ± 3.3E−05	
Note:

d, diameter at breast height (cm); h, height (m); cl, confidence limits (95%); ±, mean standard deviation.

We used the model n(t + 1) = A *n(t), where matrix A ( aij) describes how individuals in each class in the vector n(t) (structure of size) contributes to the size class n(t + 1), and specifically contains the values Pij, Gij and Fij (Table 2, Fig. 1). The data obtained from the post-breeding census allowed for an estimate of the fecundity of class >0 ≤10 cm dbh with the equation P3* F4 (where P3 is the probability of the size-class transition >0–10 cm dbh to >10–20 cm dbh, and F4 is the fecundity of size class >10–20 cm dbh; Kendall et al., 2019) (see Fig. 1). This model provides information to obtain the population stable size distribution, λ and vital rates. We estimated a confidence interval for λ for each population with Program SIM_VAR.BAS, temporal stochasticity (Ebert, 1999). The confidence limits include the three matrices by each population and the Bootstrap process was used to choose any of the three transition matrices to estimate λ with 1,000 iterations. The average of the 25 lowest and 25 highest values of λ defined 95% of the confidence limits (Ebert, 1999).

To test the demographic compensation hypothesis in relation to the asymptotic population growth rates, we proceeded by the following method: (a) taking into account the three calculated λ from each population, one thousand λs we randomly selected within each population by way of the Bootstrap process. We used thousand times the value of 1, a population in equilibrium, as population control to compare the lambdas among populations (total n = 4,000); (b) we used SAS General Lineal Model procedure (SAS, 2016), to conducted a ANOVA and multiple comparisons through Bonferroni correction; (c) as indicated in the hypothesis, no significant differences or differences greater than 1 we expected with the population control.

Elasticity analysis

A method for comparing the relative contributions to λ at different stages of the life-history is demographic loop analysis (Wardle, 1998; De Kroon, Van Groenendael & Ehrlén, 2000). We used elasticity analysis to indicates the proportional contribution (eij) to λ of the values of the transition matrix (Pij, Gij and Fij) (Fig. 1), that is, eij=(aij/λ)sij, where aij are respective transition matrix values (A) and sij=(∂λ∂aij) (Fig. 1). The elasticity and sensitivities of λ to all aij are known as E (elasticity) and S (sensitivity) matrices, respectively (Caswell, 2001). The sum of the eij values from the matrix E is 1. Elasticity analysis is used to evaluate the relative weight of each matrix elements (transitions or survival in size classes in matrix A) and to select those that are more relevant to sustainable management and conservation (e.g., Link & Doherty, 2002). We calculated the average elasticity matrix for each population using their corresponding average transition matrix.

Contrary to population growth rates, significant differences are expected among each of the eij average values of the three elasticity matrices of each population, which would prove the existence of demographic compensation. To establish such differences, we observed the following procedure: (a) to compare each of the eij average values among populations, the three elasticity matrices of each population were used. We randomly selected one thousand values through a Bootstrap process for each of the three respective values from the three elasticity matrices per population; (b) We compared each set of one thousand values from each population to the others by way of an ANOVA analysis through the SAS GLM procedure (SAS, 2016), and multiple comparisons we conducted with the Bonferroni adjustment; (c) according to our hypothesis, we expected significant differences among each one of the respective eij average values from the three populations.

Life table response experiments (LTRE) analysis

This is an expansion of the sensitivity analysis, which includes vital rate-specific changes or variations (Mills & Lindberg, 2002). We used the LTRE analysis to understand how variations in vital rates translate into variations in the growth rates of populations under “long-term study”. Such variations, along the elasticity matrix, can be used in decision-making processes over the conservation of a given population (Ehrlén & Van Groenendael, 1998; Sánchez-Velásquez & Pineda-López, 2010). In this sense, the LTRE analysis shows the response to changes that are proportional to past variations in the transition rate, whereas the elasticity analysis predicts the response to changes that are proportional to average past transition rates (Ehrlén & Van Groenendael, 1998; Caswell, 2001). We calculated one LTRE matrix per population with the following equation:

LTREij=Vij×δλ/δaij|mean, where Vij is the standard variation of the transition index, and δλ/δaij|mean is the sensitivity of the average matrix (with n = 3) (Ehrlén & Van Groenendael, 1998).

To test the hypothesis of the presence of demographic compensation in the populations subjected to the LTRE analysis, we proceeded by the following method: (a) from each one of the three LTRE matrices (one for each population) one thousand values were randomly selected by way of the Bootstrap process (total n = 3,000 data); (b) We performed an ANOVA analysis through the SAS GLM process (SAS, 2016), and multiple comparisons being subjected to the Bonferroni adjustment. This is a robust analysis, which included the variations of all values from each LTRE matrix; (c) as enunciated in the hypothesis, significant differences are expected to be found among average values of the LTRE matrices of the populations.

The basic information and the analyses suggested by Gascoigne et al. (2023) to standardize the information obtained from demographic studies are provided both in the text and the accompanying tables.

Results

Population structures, transition matrices and λ of Quercus oleoides

In the three populations, productive individuals were those with a dbh of >10 cm, and no seed survival was there from 1 year to another (Fig. 1). In the three populations, the production of seeds occurred in each year, and an increase in the number of adults through time was observed (Fig. 2). The number of individuals (>0 cm dbh) per ha varied from 1,321 to 2,605 (Table 1), and the average number of seed produced per size class ranged were 156–571, 1,000–5,000, 8,000–10,000 and 6,500–8,000, for the size classes (dbh) >0.10, >10–20, >20–30, and >30, respectively (Appendix A, Table 1). The asymptotic population growth rates of the average matrices are higher than 1 indicates that the populations tend to increase (Table 2). The confidence intervals of λ were greater than 1 (Table 2). And among the annual transition matrix, Mesa de Veinticuatro had an exceptional λ of less than 1 (0.996) in 2017–2018 (Table 2, Appendix A). Significant differences were found in the average values of λ (from total n = 1,000 in each population versus control λ = 1) (F = 1,971.42, P < 0.0001), all populations were significantly higher than the control (λ = 1) (in all cases P < 0.0001, Miramar λ¯ = 1.076 ± 0.0006, average ± mean standard deviation; Mesa de Veinticuatro λ¯ = 1.017 ± 0.0006; and Otates λ¯ = 1.042 ± 0.0004).

Figure 2 Size structure by development stage and year of Quercus oleoides at the three locations under study, Veracruz, Mexico.

Miramar with moderate cattle ranging, Mesa de Veinticuatro no cattle ranging, Otates intensive cattle ranging. d, diameter at breast height (cm); h, high (m).

Elasticity analysis

The size classes that contributed the most to asymptotic population growth rates (λ) were the survival of individuals from >0–10 to >30 cm of dbh in Miramar (71% in their λ) (Table 3A), >30 dbh in Mesa de Veinticuatro (90.6% in their λ) (Table 3B), and >0–10, >10–20 and >30 cm dbh in Otates (84.2% in their λ) (Table 3C). Significant differences were found among the respective eij average values of each element of the elasticity matrix from populations (values being from F = 791.1 to F = 12,953.2, and for all cases P < 0.0001) (all averages and each value’s ± mean standard deviation to [n = 39] are shown in Table 3).

Table 3 Elasticity matrices means of Quercus oleoide population from (a) Miramar, (b) Mesa de Veinticuatro and (c) Otates (localities, Veracruz, Mexico.

Different letters indicate significant differences among populations (elements of the matrices, p < 0.0001).

Size	Seeds	<1.3 h	>0–10 d	>10–20 d	>20–30 d	>30 d	
(a) Miramar	
Seeds	0.000	0.000	0.002a ± 0.001	0.006a ± 0.002	0.019a ± 0.003	0.027a ± 0.010	
<1.3	0.054a ± 0.004	0.007a ± 0.004	0.000	0.000	0.000	0.000	
>0–10	0.000	0.054a ± 0.014	0.132a ± 0.028	0.0000	0.000	0.000	
>10–20	0.000	0.000	0.052a ± 0.014	0.112a ± 0.034	0.000	0.000	
>20–30	0.000	0.000	0.000	0.046a ± 0.013	0.175a ± 0.106	0.000	
>30	0.000	0.000	0.000	0.000	0.027a ± 0.010	0.287a ± 0.031	
(b) Mesa de Veinticuatro	
Seeds	0.000	0.000	3E−5b ± 8E−5	0.0001b ± 0.0003	0.0004b ± 0.002	0.007b ± 0.004	
<1.3	0.007b ± 0.005	0.001b ± 0.001	0.000	0.000	0.000	0.000	
>0–10	0.000	0.007b ± 0.005	0.008b ± 0.015	0.000	0.000	0.000	
>10–20	0.000	0.000	0.007b ± 0.004	0.013b ± 0.020	0.000	0.000	
>20–30	0.000	0.000	0.000	0.007b ± 0.004	0.031b ± 0.091	0.000	
>30	0.000	0.000	0.000	0.000	0.007b ± 0.004	0.906b ± 0.115	
(c) Otates	
Seeds	0.000	0.000	0.002c ± 5E−5	0.031c ± 0.007	0.001c ± 0.001	0.003c ± 0.002	
<1.3	0.036c ± 0.004	0.007c ± 0.036	0.000	0.000	0.000	0.000	
>0–10	0.000	0.036c ± 0.004	0.115c ± 0.072	0.000	0.000	0.000	
>10–20	0.000	0.000	0.035c ± 0.004	0.537c ± 0.022	0.000	0.000	
>20–30	0.000	0.000	0.000	0.004c ± 0.004	0.002c ± 0.022	0.000	
>30	0.000	0.000	0.000	0.000	0.003c ± 0.002	0.190c ± 0.059	
Note:

d, diameter at breast height (cm); h, height (m); ±, mean standard deviation.

LTRE analysis

Transition and survival variations in LTRE matrices also differed across populations. In Miramar, the highest variations occurred in transitions and survival to >10–20 and >20–30 cm dbh (Table 4A). In Mesa de Veinticuatro, the highest variation was in the survival and transition of >10–20 cm dbh (Table 4B). On the other hand, the maximum variation in Otates occurred in seed production among individuals of the class >10–20 cm dbh, as well as in the transitions and survival from <1.3 m high (Table 4C). Significant differences were found among the values from LTRE matrices of the populations (F = 204.64, P < 0.0001). LTRE average and mean standard deviation are 0.0062 ± 0.002, 0.0038 ± 0.0012 and 0.0055 ± 0.0019, to Miramar, Mesa de Veinticuatro, and Otales, respectively.

Table 4 Life-table response experiment of three Quercus oleoide populations.

Different superscript letters in a population’s name indicate significant differences (p < 0.0001) among the set of elements of the matrices.

Size	Seeds	<1.3 h	>0–10 d	>10–20 d	>20–30 d	>30 d	
(a) Miramara	
Seeds	0	0	0.0017	0.0045	0.005	0.005	
<1.3	0.041	0.0054	0	0	0	0	
>0–10	0	0.00312	0.00353	0	0	0	
>10–20	0	0	0.02748	0.02881	0	0	
>20–30	0	0	0	0.03357	0.03363	0	
>30	0	0	0	0	0.02917	0.00014	
(b) Mesa de Veinticuatrob	
Seeds	0	0	0.00073	0.00042	0.00173	0.01245	
<1.3	0.01592	0.00106	0	0	0	0	
>0–10	0	0.00327	0.01812	0	0	0	
>10–20	0	0	0.03096	0.01081	0	0	
>20–30	0	0	0	0.01175	0.01496	0	
>30	0	0	0	0	0.01512	0	
(c) Otatesc	
Seeds	0	0	0.00424	0.0404	0.00293	0.00212	
<1.3	0.03676	0.02896	0	0	0	0	
>0–10	0	0.03936	0.015565	0	0	0	
>10–20	0	0	0.00567	0.01045	0	0	
>20–30	0	0	0	0.00492	0.00673	0	
>30	0	0	0	0	0.00065	2.53E−06	
Note:

d, diameter at breast height (cm); h, height (m).

Discussion

Population structure

Population size varies greatly among altitudes and years (Fig. 2). Due to the presence of livestock in two of the populations (moderate livestock farming in Miramar and intense livestock farming in Mesa de Veinticuatro), a lower regeneration of oaks would be expected. However, there is evidence that oak regeneration can be favored by the presence of livestock and ungulates in general, not only because some oaks are not palatable to livestock due to their toxicity (tannic acid) and unpleasant taste (e.g., Q. intricata Trel., Q. pringlei Seemen, Bacarrillo Rangel, 2015), but also because livestock can eliminate grass and herbs, which favors the regeneration of oaks (Hernández-Vargas et al., 2000; Bobiec et al., 2011; Leal, Bugalho & Palmeirim, 2022). There is little consensus as to whether ungulates in general have a positive or negative effect on oak regeneration. Intervening factors are the number of livestock individuals, the presence and amount of forage grasses and herbs, and the grazing season for livestock and other wild ungulates (Leal, Bugalho & Palmeirim, 2022). In any case, maintaining a steady population growth rate is more important than having a high regeneration potential (Tyler, Khun & Davis, 2006).

Also, given the ability of Q. oleoides to regenerate under its own canopy (tolerant to shade) (Vanclay, 1994), we expected to obtain an inverted J-shape size distribution as other Quercus populations (Duan et al., 2024; Lian et al., 2024), mostly in Mesa de Veinticuatro (with no cattle ranging), that is, a larger number of individuals in each preceding size class. However, the distribution of the number of individuals within size classes at the three different altitudinal levels (Fig. 2) suggests regeneration pulses (Martínez-Ramos & Álvarez-Buylla, 1995), that similarity occur in others Quercus (Jazib, 2023; Šimková et al., 2023; Duan et al., 2024). This rather common phenomenon among tree species can be attributed either to seed synchronicity or to disturbances that detonate massive germination processes (Janzen, 1971). In 2019, a decrease in regeneration was observed in all three populations (individuals <1.30 m high) (Fig. 2). Comparatively, in the Himalayas, Q. semecarpifolia was found to have very scarce regeneration at intermediate altitudes and null at lower altitudes (Fartyal et al., 2022). A similar pattern was evident towards the end of our study, with a drastic decrease of regeneration at an intermediate altitude (Fig. 2). A scarce regeneration has also been reported in multiple studies on the California oak woods, which has nevertheless been enough to ensure the survival of populations (Tyler, Khun & Davis, 2006). In elucidating the stability of populations in general, population growth rates are more reliable indicators than simple regeneration observation (Tyler, Khun & Davis, 2006).

Demographic compensation

Demographic compensation can be observed in the asymptotic population growth rates. Located at different altitudes and having a different disturbance history, the three populations under study show significant differences in terms of control (λ = 1). In all cases, λ > 1.0 could change in response to density-dependence effects until reaching a λ ≈ 1 in the not-so-distant future. Moreover, there is evidence that compensation can follow different paths depending on the adjustments made to the different vital rates. In our case, this is evidenced in the elasticity and the variation (LTRE) matrices. This suggests the presence of different selection pressures among Q. oleoides populations (Van Groenendael, De Kroon & Tuljapurkar, 1994; De Kroon, Van Groenendael & Ehrlén, 2000) and a clear demographic compensation.

Population growth rate and management decision

Asymptotic population growth rates summarize population dynamics, and their confidence intervals indicate its variations. In our study, λs were slightly superior to 1, this coincides with the population growth rates of other oak species (Tlapa-Almonte, 2005; Alfonso-Corrado, Clark-Tapia & Mendoza, 2007; Trigueros Bañuelos, Villavicencio García & Santiago Pérez, 2014).

In the Miramar and Mesa de Veinticuatro populations, with moderate disturbance and without disturbance respectively, the size classes that contributed the most to λ were survival classes (specifically >20–30 and >30 cm dbh). However, in the Otates population the contribution was concentrated in the survival of size classes from >10–20 and >30 cm, the latter one presenting the highest disturbance (Table 3). The highest contribution to regeneration was observed in Miramar (intermediate disturbance) and the lowest in Mesa de Veinticuatro (without disturbance). This suggests that disturbances could have caused changes in the asymptotic population growth rate, and these are the most important ones for the conservation of this population. Conversely, conservation in the Miramar and Mesa de Veinticuatro populations relied on the proportionality of size classes >20–30 up to >30 cm dbh. Disturbances as natural selection processes may change the loops within the annual transition of populations (e.g., Sánchez-Velásquez et al., 2002; Elogne et al., 2023). Differences in size structure, size-class contributions (survival and transition) and elasticity values among populations of Q. oleoides with similar asymptotic population growth rates along an altitudinal gradient and different disturbance conditions suggests demographic compensation (Villellas et al., 2015; Yang et al., 2022).

Another important aspect of population dynamics and its interpretation when making management and conservation decisions is vital rate variation (e.g., Baldauf et al., 2015). As mentioned earlier, LTRE analysis provides information on transition and survival variations in a matrix. In this sense, our results also indicate a clear significant difference among the three populations in terms of variation (Table 4). When comparing the LTRE and elasticity matrices (Table 3), we concluded that the highest transition and permanence values in the LTRE matrices would merit more attention in terms of management decisions, for they are important stages in the contribution to λ and at the same time they have a wide variation (Table 4).

Other studies have obtained λ < 1, that is, population decline, notably in four oak species, two of them deciduous and the other two evergreen (Tyler, Khun & Davis, 2006; Alfonso-Corrado, Clark-Tapia & Mendoza, 2007; Conlisk et al., 2012; Trigueros Bañuelos, Villavicencio García & Santiago Pérez, 2014). This suggests the presence of effects that could surpass the demographic compensation threshold of a population, but which could also attain a balance during a long period until reaching λ ≈ 1. More detailed studies and simulations could help elucidate, on the one hand, the stages that allow demographic compensation and, on the other, identify the variations of the asymptotic population growth rates and the vital rates. It would also be beneficial to include other aspects such as environmental drivers (Römer et al., 2021) and the ability to produce sprouts after disturbances (Oldfield & Eastwood, 2007; Oldfield, Evans & Oldfield, 2021).

Finally, we have to acknowledge that the average vital rates and the asymptotic growth rates obtained were dependent on the environmental conditions prevalent during the years 2016–2019, and that they are likely to change under different conditions.

Conclusions

The three Q. oleoides populations have different abilities to transit on to demographic compensation and achieve demographic balance through different strategies in their vital rates. The behaviour of their asymptotic population growth rates (λ ≥ 1.0), the differences in the elements of their elasticity matrices, and the variations in their LTRE analyses all were according to what was expected in the hypothesis. This confirms the ability of populations to keep balance by adjusting vital rates through demographic compensation.

Limitations and recommendations

Though the balance threshold in the three populations under study remains unknown, it is important to keep in check disturbance by extractive activities. Further studies on the similarities and dissimilarities of stochiometric traits such as carbon, nitrogen, and phosphorous concentration in leaves and other tissues in relation to the soil and environment could provide the tools to measure demographic compensation capabilities. In the absence of accurate climate histories and disturbance data for each population, we had to rely on our non-quantitative field observations, such as the presence of stumps and livestock dungs, and we only knew the number of cows per ha (Table 1). Also, three years may be a short time to know the variation in regeneration pulses, due to the maturation cycles of oak trees. Furthermore, it is difficult to distinguish among disturbance impacts and altitude effects in our study design, since different number of cows per ha at each elevation. Finally, to better understand the behaviour of Quercus oleoides populations, more demographic-rate measurements would be needed.

Supplemental Information

Supplemental Information 1 Transition, Variation (LTRE) and elasticity matrices of Quercus oleoides populations of Veracruz center, Mexico.

Supplemental Information 2 Transition and permanence across three years Quercus oleoide populations.

We want to thank the four reviewers and the editors for their comments and observations, which have greatly contributed to the improvement of this manuscript, and to Luis L. Esparza for his help with the English-language version.

Additional Information and Declarations

Competing Interests

The authors declare that they have no competing interests.

Author Contributions

Carlos Flores-Romero conceived and designed the experiments, performed the experiments, analyzed the data, prepared figures and/or tables, authored or reviewed drafts of the article, and approved the final draft.

Lázaro Rafael Sánchez-Velásquez conceived and designed the experiments, performed the experiments, analyzed the data, prepared figures and/or tables, authored or reviewed drafts of the article, and approved the final draft.

Miguel Equihua performed the experiments, authored or reviewed drafts of the article, and approved the final draft.

María del Rosario Pineda López conceived and designed the experiments, prepared figures and/or tables, authored or reviewed drafts of the article, and approved the final draft.

Enrique Alarcón Gutiérrez performed the experiments, authored or reviewed drafts of the article, and approved the final draft.

Yareni Perroni conceived and designed the experiments, authored or reviewed drafts of the article, and approved the final draft.

Data Availability

The following information was supplied regarding data availability:

The raw data are available in the Supplemental Files.

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
