# Peer review of "Demographic compensation occurs in populations of Quercus oleoides Schltdl & Cham in fragments across an altitudinal gradient"

_PeerJ, doi:10.7717/peerj.18980_

## Round 0.1 · original submission · Major Revisions

Please see the comments of the 2 reviewers. And make sure (with evidence) that the concept of "demographic autopoiesis" is not simply "demographic compensation".

**Language Note:** The review process has identified that the English language must be improved. PeerJ can provide language editing services - please contact us at [email protected] for pricing (be sure to provide your manuscript number and title). Alternatively, you should make your own arrangements to improve the language quality and provide details in your response letter. – PeerJ Staff

Reviewer 1 ·

Basic reporting

The article entitled "Autopoiesis Occurs in Populations of Quercus oleoides Schltdl & Cham in Fragments of an Altitudinal Gradient" focuses on characterizing the demographic traits of the species Quercus oleoides. To achieve this, the authors established permanent sites in three fragmented populations. The authors state that their objective was to identify autopoietic responses. Overall, I find the paper quite interesting, and its content is certainly relevant to the journal PeerJ. However, it could benefit from some fine-tuning before publication.

Substantial work in writing and editing is needed, especially in the Materials and Methods and Results sections. Specifically, the authors should strive to enhance the text's fluidity and homogenize terms and expressions.

Furthermore, I noticed that while the authors frequently mention the terms "demographic parameters" or "demographic growth" in the Introduction, these terms are only briefly mentioned twice in the Materials and Methods section without clear explanation. Additionally, there is no mention of these terms in the Results section. Clarifying and ensuring consistency in the use of these terms throughout the manuscript would improve its clarity and coherence.

Experimental design

The experimental design appears to be valid. However, it would be beneficial to clarify the variables studied in a more precise manner. For instance, in the Abstract (L.34), it is mentioned that "three annual transition stages (survival, reproduction, growth, and mortality) were recorded." However, the Materials and Methods section does not clearly describe how these stages were recorded. Were measuring instruments utilized? If so, which ones? Could you provide information on the commercial brand of these instruments? Additionally, it would be helpful to present a summary of the descriptive statistics, including the total number of individuals observed (n), the units of measurement, and statistics such as minimum, maximum, median, and average, for each population studied. This information is crucial for determining if the sample was representative.

Regarding lines 158-159, it is not entirely clear to me how the spatial distribution of the five square plots in each altitudinal stratum was defined. Were they selected randomly? And how far apart were they from each other?

The Data Analysis section remains somewhat confusing. It would be beneficial to organize the content in a clearer manner, outlining the steps taken sequentially. Perhaps consider using indents or listing the steps followed. In summary, further writing refinement is required. Nevertheless, the methods of analysis seem to be appropriate.

Validity of the findings

The results may have validity but lack clarity. Two strong suggestions: (1) Try to write the results section as clearly as possible, using the words according to the context of the study. Instead of using lamda, transition matrix, etcetera, it is better to say, what is expressed or represented by lamda, transition matrix, etcetera. (2) Since the validity of the results depends largely on the quality of the data used in the field, the basic information (descriptive statistics of the data in Math & Meth) should be shown. For now, the information presented is insufficient.

Additional comments

In lines (L) 42-43, when the authors state: “may indicate a substantial phenotypic plasticity”. Phenotype? that's a good sentence, is that what they want to express?

L. 45, 46. Rather, climate varies according to the altitudinal gradient and latitude. In addition to latitude and altitude, other factors also play an important role. For example, the distance from the equator, solar radiation, longitude, mountain ranges, and so on. In this regard, it is important that the authors review and cite studies that talk about the variability of climate metrics. Namely, there are mainly two: temperature and precipitation. But cite works from the area of study, and if there are none, cite works from the country where the study is made. The current citations talk about other things, and do not refer precisely to climate and latitude.

Use simple and direct expressions to refer to things. Otherwise, reading is hindered. The text loses fluency. Replace generalist or ambiguous words and expressions. For example, "demographic parameters", "population stability" or "concomitant" etc. Instead, use concrete terms or examples. What kind of "demographic parameters", could you list these constants?
There are sentences that sound inconclusive or unclear. For example, "...making a series of the changes in the vital rates (L.162-163) ... in the vital rates of whom or what? Like what?
A strong quote is missing, stating the following: "which is explained by the displacement and adaptation undergone by this species during the last glacial periods of the Pleistocene"L.
L. 154 regarding “with no apparent cattle," since it is a stratification criterion for data analysis, the authors should clarify unambiguously: with or without pasteurizer?

L. 158. Replace “five permanent squares of 400” with “five square plots of 400 m2”

L. 160. Improve the sentence, please

L. 162… A GPS receiver instead of a GPS.

L.187-188. Better to write like this: with 1000 iterations…

The "Population Structures" section of the Results section. It has only two lines. It would be better to join the two lines with another paragraph.



The "Elasticity analysis" section should be rewritten using clear and simple language. On the other hand, it is convenient to show the audience where it can be verified. For example, when the authors state: << he stasis of individuals of size classes 0-10 and > 30 cm of dbh was a key factor in the demographic growth (λ) of the populations of Miramar and Mesa de Veinticuatro>>, cite a Table a figure, or mention a value that evidences this statement.



In results, for the authors, it might be very obvious what they themselves write, but not for their readers. So, it is necessary to clarify things.

In line 254, the authors talk about "control". What do they mean? Was there a control population? There is no mention of this in Materials and Methods.

The discussion is of good length and reads more smoothly than the results section.

L. 349-350, seems to have no connection with the paragraph preceding it. It could even be deleted.

The results presented do not clearly support the conclusions. There must be correspondence. The results should be presented in such a way that the audience comes to the unequivocal conclusion of what is stated in Conclusions

Reviewer 2 ·

Basic reporting

The topic of the paper entitled " Autopoiesis occurs in populations of Quercus oleoides Schltdl & Cham in fragments of an altitudinal gradient (#95816) " is noteworthy and falls within the scope of PeerJ. In this manuscript, the authors have analysed the impact of altitudinal variations on Quercus oleoides population using the scientific methods. The research topic and data collected are intrinsically interesting because such types of the studies impact the cause-effect relationships between soil, plants environment and society. The research topic and findings are inspiring.

Experimental design

The research topic and design falls within the scope of PeerJ.

Validity of the findings

The research topic and findings are inspiring. The set of measurements was comprehensive.

Additional comments

The following suggestions may help to improve the quality of the Ms. further. Overall, the title clearly reflects the contents. The introduction does establish the existing state of knowledge but needs minor revision as suggested. Authors should add the knowledge gap at the end of the introduction with proper citations. Do not repeat the results in the discussion and conclusions. The citations need up-dated.
English needs minor grammatical improvement.
I have made more corrections and suggestions directly in the manuscript.
Please revise the paper accordingly.

Annotated reviews are not available for download in order to protect the identity of reviewers who chose to remain anonymous.

---

## Round 0.2 · Minor Revisions

Please check that autopoiesis is indeed present and support your definition of autopoiesis with a strong citation. Also note the reviewer's concerns.

Reviewer 1 ·

Basic reporting

--

Experimental design

--

Validity of the findings

--

Additional comments

Overall, the authors have made improvements to their work, although the results section could benefit from further refinement. But given some of the terminology changes, it may be prudent for the authors to reassess whether "autopoiesis" is the most appropriate term for this paper.

"Autopoiesis" is a central term in the paper, appearing prominently even in the title. Upon a cursory examination of the concept of Autopoiesis, I discovered that it has been subject to discussion and criticism (see Luisi, 2013; Boden, 2000). For instance, Boden (2000) on page 118, argues: "In the penultimate section ("Autopoiesis, Biology, and Cognition"), I show that Maturana and Varela's unorthodox biological approach is problematic for cognitive science in general."

Considering various definitions of autopoiesis from different sources (provided below), and taking into account the authors' definition of "demographic autopoiesis" as "a complex process by which, during their life history, individuals manage to compensate for the effects of anthropogenic disturbance" (L. 84-85), it leads me to consider whether there might be more suitable terms for the concepts presented in their work.

It appears that a previous reviewer suggested that the concept of "demographic autopoiesis" discussed in the paper could be refered by the term "demographic compensation."

In this regard, I pose a question to the authors: Could "demographic compensation" as discussed in Villellas (2015) and in Yang et al. (2022) be analogous to the concept referred to in their paper?

In summary, it would be prudent for the authors to reconsider whether to persist in using the term "autopoiesis" or to explore alternative terms or expressions that might better convey their ideas without ambiguity or uncertainty.

Luisi, P. L. (2003). Autopoiesis: a review and a reappraisal. Naturwissenschaften, 90, 49-59.
Boden, M. A. (2000). Autopoiesis and life. Cognitive Science Quarterly, 1(1), 115-143.

Zeleny, M. (1981). What is autopoiesis. Autopoiesis: A theory of living organization, 4-17.

Villellas, J., Doak, D. F., García, M. B., & Morris, W. F. (2015). Demographic compensation among populations: what is it, how does it arise and what are its implications?. Ecology letters, 18(11), 1139-1152.

Reviewer 2 ·

Basic reporting

Authors have revised the paper according to the comments made by reviewers. They have answered all the comments and included all the suggestions.
I am fully satisfied with the revised version and it should be accepted for publication.

Experimental design

Appropriate

Validity of the findings

Appropriate

Additional comments

The revised paper be accepted.

---

## Round 0.3 · Major Revisions

The reviewer drew attention to some serious problems in the material and method. In addition, it was noticed that the data set of 5 x 3 400m2 is very small. The measurement period of 4 years is very short for a species that can certainly be older than 50 years. And the study was carried out in only one region, whether the species has a large distribution area. Please add more sample plots.

Reviewer 1 ·

Basic reporting
* * *
Experimental design
* * *
Validity of the findings
* * *
Additional comments

I defer to the editor's judgment regarding whether the plots used (which were not described in the first round) were appropriate and sufficient for making inferences.

It is important to note that the authors did not implement a sampling design to mitigate the effects of various sources of variation. They claim that the plots were randomly established (see Line 168).

It is challenging to believe that across an altitudinal range from 0 to 840, there is no variation in undulation, exposure, or slope (considered noise factors). In other words, the reported results could potentially be influenced by other factors or chance.

Regarding the three altitudinal levels, there should also be three types of grazing, as grazing is one of the two main criteria: null, moderate, and intensive. Without replication, the observed result could be attributed to chance or another factor.

Additionally, there are minor issues: the conclusion of the Abstract is ambiguous, as it could apply to any other species. Figure 2 could be enhanced for clarity.

---

## Round 0.4 · Major Revisions

I highly recommend not to include autopoieses as the main thrust of the paper. We have many areas of research within population ecology that the authors are not consulting. I fear the attribution of autopoieses dismisses and obfuscates known research.

· Appeal

Appeal

It is unfortunate that you do not have an argument to reject our article and only make a speculative opinion. In the previous response that we sent you (enclosed), there is our argument (and evidence from articles published in prestigious scientific journals with a methodology very similar to ours).

We really felt discriminated against, we showed you that you and the referee were not right in their last observations and perhaps that affected your decision. Your "argumentation", with which we do not agree, would have been made by you and the arbitrator from the beginning of the review process. It is unethical to make "arguments" without substance and at the wrong time.

We respect your partial decision, but we do not agree.


· · Academic Editor

Reject

Unfortunately, the reviewer's concerns and my comments were not taken into account. The reviewer pointed out some serious problems in the material and the method. It was also noted that the data set of 5 x 3 400m2 is very small. The measurement period of 4 years is very short for a species that may well be older than 50 years. And the study was only carried out in one region, although the species has a large distribution area. We are sorry that these problems were first uncovered in the last round. In addition, there are still errors in the labeling of the tables (e.g., Table 1: I am missing values of “Tma maximum temperature, Tmm medium temperature, Tmi minimum temperature” in the Table). In Table 1 the Mean is greater than Max.

·

Basic reporting

This is my first time reviewing this manuscript, and I just want to point our that I sincerely appreciate the effort the authors have gone to in order to collect these demographic data. This contribution is, in and of itself, major. However, I have some concerns regarding the manuscript in its current form. First, with regard to autopoiesis and second with regard to the writing.

The authors present their case study as an example of autopoieses. However, despite attempts to define demographic autopoieses on lines 24, 71 and 93, I, as a reader reasonable-versed in this literature, am left a bit confused. My initial thoughts were "how does this description of autopoieses differ from demographic compensation?" After the reading the clarification on lines 86-97, I am still quite confused. In my interpretation, demographic compensation arises when vital rates negatively covary across and environmental gradient. I do not see how this differs from autopoieses - as the authors describe it. Further more, I do not see how autopoieses differs from the realities of population. For example, the authors use MPMs which are discrete time stage-structured models that project a population (n_t) across time steps n_t+1 = A_t x n_t. Due to the ergodic properties of Markov processes, this system will naturally converge on a stable structure (w) and population growth rate (lambda). This behaviour has been long-understood since the times of Leslie and Lefkovitch. Personally, I do not see the value in redefining an understood "self-regulation" dynamic as autopoieses when there is a fast literature studying these phenomena (see erogodic theory and transient dynamics for example). I think this concern was cemented when reading first paragraph of the discussion when the Miramar population was described as growing (lambda > 1) but "it could change until reaching a lambda ~ 1 in the not-so-distant future due to the effect of autopoiesis." Whilst I agree the lambda cannot stay above 1 ad infinitum, the source of this change is density-dependence which is a property of the environment - not the population in question.

My second concern is with regard to the writing. Across the manuscript, there are typos (e.g.,67, 140, 207, 227). Secondly, the writing structure is a little hard to follow. For example, the first paragraph in the introduction is 32 lines in length. This, as a reader, is a little tricky. Ideally, each paragraph would be a bit more concise and self-contained. Furthermore, there were times when reading the manuscript where I was deeply confused. For example, line 234 is quite a weird way to complicate something simple. I had to re-read this three times. I would highly recommend going through each sentence and removing unnecessary clauses/examples - it would really help the reader.

Experimental design

no comment

Validity of the findings

I think there could be a bit more description with regard to how the findings were ascertained. First with regard to the matrices, are they from a pre-breeding or post-breeding census? Second, does it take two-years to go from adult to sapling. Right now it takes two years, is this a mistake - this is a common mistake, see Bruce Kendall's work on this https://www.sciencedirect.com/science/article/pii/S0304380019301085. Third, why did you take the geometric mean of the matrices? Whilst I understand this for population growth rates, much theory and comparative work is dependent on the arithmetic mean of said matrices. Four, more of a comment.......only vital rates are rates out of the ones mentioned on L.73. Five, do you think three annual transition matrices sufficiently captures the temporal variation in demographic rates in your species? I was surprised not to see this mentioned as a limitation.

Additional comments

Last but not least, I think a lot more can be done with this work. I was pleased to see the authors also collected stage distributions. This is vital for transient dynamics studies. Are the stage structures at equilibrium? How far are they from equilibrium? What does the transient portfolio of these populations look like? All of this and more can be done with these data - I was hoping to see this as a reader.

And finally, I don't like it when reviewers do this but I would recommend reading a paper I put together with a lot of population/comparative demographers where we outline how best to communicate the information stored in papers such as these: https://besjournals.onlinelibrary.wiley.com/doi/full/10.1111/2041-210X.14164

---

## Round 0.5 · Major Revisions

Unfortunately, the prior reviewer could not be reached for a review, so a new reviewer had to be brought in. Please answer the new reviewer's questions and comments.

·

Basic reporting

--This is my first time reviewing this manuscript, and overall I think the subject and methodology are strong and would be a valuable contribution to this journal. That being said, my main recommendations are (1) to revise the writing throughout to avoid repetition and awkward phrasing, (2) to present more detailed results on the data you collected, (3) to clarify some of your methods and improve the figures, and (4) to do more detailed interpretation of these data in your discussion and recommendations.

--There is awkward sentence phrasing throughout the methods. I suggest trying to use active voice (“We measured x..”_ rather than passive voice (“X was measured...”) to make these statements more direct and therefore a bit clearer.

--The first two sentences in the results section are also pretty awkwardly phrased.

--The discussion is very repetitive, and I wanted much more detail and interpretation of the results. You repeatedly state that you found the presence of demographic compensation in each population (lines 285-289 state this in three separate sentences, line 293 states it again, again on lines 313-315, lines 325-328, lines 330-333, again in lines 341-345, lines 356-358, and then again in the conclusions at the end). If you add this up, it means that almost half of the discussion is dedicated to saying that the population growth rates were greater than 1 and therefore demonstate demographic compensation. It would be more interesting to either investigate more what the differences between the three populations were, or to talk in more detail about what these differences or these findings might mean for the conservation and management of these populations.

--I’m assuming that you are not native English speakers based on the writing here, and so you may want to run some of the text through an AI tool to fix the grammar and sentence structures here if you don’t have access to a native English speaker who could help edit the writing style. Notebook LM is a Google AI tool that will not use your inputs for future training, and is also free, but of course there are other options too.

Experimental design

--5 plots per elevation for total of 15 plots? You should state this clearly at the beginning

--you counted all acorns in each tree in each plot? Or did you do a subset of trees in each size class? What time of year did you do the acorn counts? Did you do this at the same time for each population? What about acorns that had already fallen?

--for your matrices, were you using means from across the 5 plots? Or combining the 5 plots for counts? I was confused throughout this on whether you were analyzing means for the 5 plots or combining the data. And for acorns, are you using means per tree? Per plot? This needs to be clarified.

Validity of the findings

--It would be very helpful to report some basic results before you get into your matrices – what was the mean density of trees per zone, for example, or what was the mean and range in DBH? What was the mean and range in acorn production or in seedling density? How different were these values across the three populations?

--since you measured each individual tree for three years, presumably you also have DBH growth rates on the individual tree level? Did you think about including these in the analyses? That’s really great data to have collected, and it seems a shame not to do anything with it.
--relatedly, how did you ensure consistency in DBH measurements from year to year? Did you have the same people doing the measures? Did you mark 1.3 m on the trees? There can be a lot of variation from person to person measuring even the exact same tree at the same time.

--why did you use a log scale for the size class figure? This really obscures the shape of the population, and it would be helpful to see the actual values. Also, even in your ln scale usually the y-axis should show consistent increments from one tick mark to the next, but you have us going from 1 to 3 to 9 to 27? But those are the exponents, not the values themselves, correct? That makes it almost impossible to visually interpret what is going on here. I’m guessing that the problem is the number of acorns is dramatically higher than the number of seedlings, which is to be expected. In which case, I would make a figure just of the actual size classes (from seedlings to large trees), and then report the number of acorns in either a separate figure or table. You should also color code the years or put some marking to indicate year on the figure itself rather than simply stating it in the figure title.

--Table 3 – are the letters indicating significant differences looking at differences between populations or differences from one size-class to another? Can you clarify this?

--Table 1 – it would be nice to add mean (and maybe max) DBH values in here as well. Also mean number of seedlings <1.3 m in height.

--for the acorns, do you have mean acorns per tree somewhere? Mean acorns per ha? It would be interesting to see these reported. And also to see what the relationship was between number of acorns and number of new individuals recruiting into the <1.3 m height size-class was? In other words, what percentage of acorns seemed to actually recruit into a seedling? This would be a valuable result to report on its own.

-in the limitations you note that you recorded presence of stumps and cattle faeces, but did you do any analyses with these? That would be also really interesting to report and analyze.

--I would add in the limitations section that 3 years is a really short time for trees which can live for well over 100 years. The pulses in regeneration due to masting cycles in oaks can be on much larger timeframes than three years, so you wouldn’t be likely to capture that type of variation, although you do mention it when discussing the population structures you find. You should also mention that it is difficult to distinguish between the impacts of elevation and disturbance in your study design, since the disturbance levels are very different at each elevation.

---

## Round 0.6 · Minor Revisions

There is huge improvement in the manuscript. Only minor revisions are needed.

·

Basic reporting

There is huge improvement in the manuscript in the writing and beyond. Suggestions for writing that needs minor improvements are below:

Line 20 – “ability to transit on to” seems to be a typo, or is a strange word choice. Is this supposed to be “transition to”?
Introduction is much improved
Line 153 – “corns” should be “acorns”
Line 241—“no seed survival from one time step to another” – what does this mean?
Results also much improved
Line 279 – “floors” should be “levels”
Discussion—you may want to consider subheadings here for “4.1 Population structure”, “4.2 Demographic compensation” and so forth to help orient the reader to which results you are discussing.

Experimental design

No comment

Validity of the findings

Overall this manuscript is greatly improved, and I recommend only minor revisions. In terms of content and interpretation, my only comments are below:

Lines 278 – 291 – you seem to be attributing all differences to elevation when there are also very meaningful differences in disturbance in terms of the amount of grazing and also the size of the forest fragments. I think it would be helpful here to also state what the differences are instead of just stating that there are differences. In other words, the lowest elevation site with cattle grazing showed the distribution closest to an inverse J, while the middle elevation with no grazing showed a humped distribution with much higher densities in larger size class, and the high elevation showed something closer to an inverse J shape. It seems worth noting that the two areas that have cattle grazing actually have more inverse-J shaped distributions, which is somewhat counter-intuitive to me (I would have thought that cattle might eat or trample the seedlings, leading to lower densities in small size-classes). So I think they actual patterns are worth discussing and interpreting, even if you can’t prove any causation, rather than just stating that they are different. Additionally, would you expect differences here in rainfall or climate with elevation? Which site would be driest vs. wettest? That would also be worth highlighting here in the interpretation of the results.
Lines 305-318 – good job here adding in more interpretation based on disturbance. This addresses some of what I discuss above, but I still think you could add a little more to that section.

Additional comments

Congratulations on making the revisions suggested, and I am glad that they were helpful in revising the manuscript.

Meredith Martin

---

## Round 0.7 · Minor Revisions

Please complete the captions of the tables and figures. All contents of the tables and figures must be understood without reading the text of the article. In Table 2 the explanation of +- is missing. Which ci, 95% or 99%? What is “LTRE”? The units of h and d are missing in Figure 1 and 2 etc. And in the text of the manuscript: What is “SDM”? The method for calculating “ci” cannot be found.

·

Basic reporting

I reviewed the author response and do not need to review this article again.

Experimental design

I reviewed the author response and do not need to review this article again.

Validity of the findings

I reviewed the author response and do not need to review this article again.

Additional comments

I reviewed the author response and do not need to review this article again.

---

## Round 0.8 · accepted · Accept

I think that the manuscript has now been improved to the level required for publication.